# Quantifying the financial burden of households' out-of-pocket payments on medicines in India: a repeated cross-sectional analysis of National Sample Survey data, 1994–2014

Sakthivel Selvaraj, Habib Hasan Farooqui, Anup Karan

Public Health Foundation of India, New Delhi, India

**Correspondence to**
Dr Habib Hasan Farooqui;
drhabibhasan@gmail.com

## ABSTRACT

**Objective** The objective of this research is to generate new evidence on financial implications of medicines out-of-pocket (OOP) payments for households. Another objective is to investigate which disease conditions contributed to a significant proportion of households' financial burden.

**Setting** All Indian states including union territories, 1993–2014.

**Design** Repeated cross-sectional household surveys.

**Data** Secondary data of nationwide Consumer Expenditure Surveys for the years 1993–1994, 2004–2005 and 2011–2012 and one wave of Social Consumption: Health for the year 2014 from National Sample Survey Organisation.

**Outcome measures** OOP expenditure on healthcare in general and medicines in specific.

**Results** Total OOP payments and medicines OOP payments were estimated to be 6.77% (95% CI 6.70% to 6.84%) and 4.49% (95% CI 4.45% to 4.54%) of total consumption expenditure, respectively, in the year 2011–2012 which marked significant increase since 1993–1994. These proportions were 11.46% (95% CI 11.36% to 11.56%) and 7.60% (95% CI 7.54% to 7.67%) of non-food expenditure, respectively, in the same year. Total OOP payments and medicines OOP payments were catastrophic for 17.9% (95% CI 17.7% to 18.2%) and 11.2% (95% CI 11.0% to 11.4%) households, respectively, in 2011–2012 at the 10% of total consumption expenditure threshold, implying 29 million households incurred catastrophic OOP payments in the year 2011–2012. Further, medicines OOP payments pushed 3.09% (95% CI 2.99% to 3.20%), implying 38 million persons into poverty in the year 2011–2012. Among the leading cause of diseases that caused significant OOP payments are cancers, injuries, cardiovascular diseases, genitourinary conditions and mental disorders.

**Conclusions** Purchase of medicines constitutes the single largest component of the total OOP payments by households. Hence, strengthening government intervention in providing medicines free in public healthcare facilities has the potential to considerably reduce medicine-related spending and total OOP payments of households and reduction in OOP-induced poverty.

### Strengths and limitations of this study

► The study used multiple points of time and nationally representative data set to highlight the financial burden of households out-of-pocket (OOP) payments on medicines in India.
► The paper links medicines OOP payments by households with leading disease conditions and identify key disease conditions which cause medicines OOP payments.
► The study has limitations as it uses arbitrary threshold for measuring catastrophic payments.
► The ailments, disease conditions and the associated OOP expenditure reported by the households in the survey are self-reports and not clinically diagnosed.

## BACKGROUND

Households' in India bear significant financial burden on account of medical treatment, as the current prepayment and risk pooling mechanisms are inadequate. Since both government funding and social health insurance contributions are insufficient to meet healthcare needs of households, over three-fourth of all healthcare payments are paid out of pocket (OOP) at the point of service delivery while medicines purchase (approximately 63%) account for the single largest component of these payments.[1] Available literature suggests that medicine OOP spending has dominated total OOP payments over the years.[2] Hence, it can be suggested that expenditure on medicines is major cause of catastrophe and impoverishment at the household level.[3]

India has the distinction of being pharmacy of the global South—supplies affordable, life saving, quality generic medicines. It ranks 4th in terms of volumes and 13th in terms of value of pharmaceuticals produced globally.[4] However, according to a WHO report,

around 68% of the Indian population have limited or no access to essential medicines.[5] In addition, literature suggests that over last two decades availability of free medicines in public health facilities has declined from 31.2% to 8.9% for inpatient care and from 17.8% to 5.9% for outpatient care.[6] Another study demonstrated that medicines purchase alone constituted over 70% of overall OOP payments. In addition, the study demonstrated that by removing OOP payments for outpatient care on medicines, the percentage of people falling below poverty because of spending on health reduced to just 0.5% whereas removing OOP payments for inpatient care resulted in a negligible decline in poverty head count ratio and poverty gap highlighting the role of medicines expenditure in healthcare-related impoverishment.[7]

Using impoverishment tool to measure affordability, one study assessed the impoverishment effect of medicines purchases by households in 16 low-income and middle-income economies.[8] Comparing four key medicine prices to household income, and using World Bank poverty levels of US$1.25 or US$2 PPP per day, the study concluded that a substantial number of people had to bear financial burden due to unaffordability of medicines. For example, it was pointed out that an originator brand atenolol purchase by individuals would push an additional 22% of population below the US$1.25 PPP per day measurement while even a generic equivalent of atenolol was likely to push about 7% of population below poverty levels in Philippines.[9] Analysing economic implications of non-communicable disease (NCD) in India, a few studies also reported in the past that households incur significant OOP payment burden in certain conditions like cardiovascular diseases (CVDs) and cancers.[10 11] Using 2004 National Sample Survey Organisation (NSSO) data, another study highlighted that hospitalisation with CVD resulted in 12% higher odds of incurring catastrophic spending and 37% greater odds of falling into poverty compared with those hospitalised with communicable conditions. For cancer, the impact was greatest with the odds of catastrophic expenditures 170% higher than the odds of incurring catastrophic spending when hospital stays are due to a communicable disease condition.[12] However, these studies do not reflect on the relative contribution of medicine in total OOP burden for the diseases they analysed.

Available evidence, both global and Indian, provides insights about the incidence of catastrophic payments and impoverishment impact of rising households OOP payments.[2 3 13–15] The literature on equity dimensions involving both catastrophe and impoverishment has attempted to address complex methodological and statistical approaches in measurements. However, there is lack of evidence on catastrophe and impoverishment on account of household's medicine OOP expenditure from inpatient and outpatient treatment costs perspective and from the disease-specific dimension. Given that medicines contribute to more than 66.6% of OOP healthcare expenditure, the focus of this research is to explore the

consequences of high medicines OOP spending at the household level. Further, we investigated which disease conditions are contributing to high financial burden on households. We attempted to answer—what is the relative burden of medicines OOP payments by households in total OOP payments, catastrophic and poverty head counts? And which disease conditions cause a relatively higher financial disruption in the living status of households?

## MATERIALS AND METHODS
### Data
The study uses secondary data from three waves of nationally representative 'Consumer Expenditure Surveys' (CES): 1993–1994, 2004–2005 and 2011–2012, conducted by the NSSO. In addition, Health and Morbidity Survey (HMS) 2014 of the NSSO was used for disease-wise distribution of OOP payment burden. While the sample size of CES varied between 100 000 and 125 000 thousand households across different rounds, the sample size in HMS 2014 was approximately 72 000 households.

The CES collect socioeconomic and demographic information of households with key focus on household spending involving roughly 350 food and non-food items. OOP medical expenses incurred by households are separately recorded for inpatient and outpatient services. The recall periods are 1 year and 30 days for inpatient and outpatient expenses, respectively. HMS collects detailed information on morbidity pattern, utilisation of healthcare services and associated expenditure by households. The HMS too separately records expenditure for inpatient and outpatient. However, unlike in CES the recall period for outpatient in HMS is 15 days.

Both CES and HMS are repeated cross-sectional surveys that are representative at the national and state levels. All districts of a state are included for sampling purposes. Households in CES are sampled evenly in quarterly subrounds beginning on 1 July and ending on 30 June of the following year, with equal numbers of households allotted in each quarterly subround, to address seasonality. In HMS, survey was completed in two subrounds during 1 January 2014 to 30 June 2014. All estimates in the present paper are sample weighted.

### Outcome indicators
Using CES, we estimated four household-level indicators involving financial burden of illness: (1) per household member monthly OOP spending on medicines (inflation adjusted), (2) OOP spending on medicines as a share of total household and non-food spending (3) percentage of households reporting catastrophic payments on medicines and (4) percentage of households slipping into poverty after netting out medicines OOP payments from households' total consumption expenditure. Total OOP spending of households was estimated by adding together expenditure involving different components of OOP payments. For inpatient OOP payment, we considered

institutional spending on medicines, X-ray, ECG, patho-logical tests etc, doctor's/surgeon's fee, hospital and nursing charges and other medical expenses. For outpa-tient OOP payment, the components of expenditure are medicines, X-ray, ECG, pathological tests etc, doctor's/surgeon's fee, family planning services and other medical expenses. Expenditure on medicines is directly reported in the data set, both for inpatient and outpatient services (see online supplementary tables A–I). All the anal-yses report mean OOP spending on two parallel tracks: aggregated (across components of OOP) OOP payments (henceforth referred to as 'total OOP') and OOP payments only on account of medicine purchase (hence-forth referred to as medicine/drug OOP).

Catastrophic payment for healthcare is defined as OOP payments being higher to a predefined threshold of total household consumption expenditure or alternatively household's non-food expenditure. For measuring cata-strophic expenditure,[2 9] instead of sticking to a particular threshold, we considered a range of thresholds.[2 16–18] We also considered alternative thresholds of OOP spending as a share of household non-food expenditure. For OOP payment induced poverty estimates, we used two different poverty lines: (1) Indian official state-specific rural and urban poverty lines[19] and (2) international poverty line based on US$1 per day per person adjusted to US$1.9 purchasing power parity (PPP) per day per person for the year 2011–2012.[20] Yet another important poverty indicator, which provides estimates around magnitude of poverty deepening, is poverty gap. Using both the poverty lines separately, we also estimated mean poverty gaps for the poor. Details of the method used for catastrophic and poverty estimates are presented in (online supplementary annexure).

In addition, we used NSSO 2014 HMS data for estimating disease level total and medicine OOP spending separately for inpatient and outpatient. Unlike CES, OOP spending in HMS 2014 has not been recorded as a part of the total household consumption expenditure and instead of esti-mating disease-wise catastrophic head count, we present distribution of disease conditions based on incidence of occurrence and range of OOP spending separately for outpatient (15 days recall converted for 30 days) and inpa-tient (365 days recall converted for 30 days). This helped identifying disease conditions, separately for outpatient and inpatient, which are high-frequency occurrence and greater incidence of OOP spending.

## RESULTS

First, we present basic financial burden indicators for the years 1993–1994, 2004–2005 and 2011–2012 in table 1. Over 80% of populations are reportedly spending OOP while seeking treatment, during 2011–2012. The proportion of population reporting any OOP payments have increased sharply from about 60% during 1993–1994 to 80% in 2011–2012. In respect to medicines spending, approximately every OOP spending is associ-ated with expenditure on medicines. There was a signif-icant increase (more than 50%) in household's total

**Table 1** Financial burden indicators, India, 1993–1994, 2004–2005 and 2011–2012

| Financial burden indicators | 1993–1994 | 2004–2005 | 2011–2012 |
|---|---|---|---|
| Percentage households reporting OOP payments | | | |
| Any OOP payments (%) | 59.2 (58.9 to 59.5) | 64.4 (64.2 to 64.7) | 80.5 (80.2 to 80.7) |
| Medicines OOP payments (%) | 57.5 (57.3 to 57.8) | 63.6 (63.3 to 63.8) | 79.0 (78.8 to 79.3) |
| Monthly per capita expenditure (INR at constant 1999–2000 prices*) | | | |
| Household consumption expenditure | 517 (515 to 519) | 619 (616 to 622) | 794 (790 to 799) |
| OOP expenditure on health | 25.59 (24.61 to 26.25) | 36.3 (35.7 to 37.0) | 54.3 (53.3 to 55.3) |
| Medicine OOP expenditure | 20.86 (19.50 to 21.25) | 26.0 (25.6 to 26.4) | 36.1 (35.5 to 36.8) |
| Share of health to total household expenditure (%) | | | |
| Share of total OOP expenditure to total household expenditure (%) | 4.84 (4.78 to 4.91) | 5.78 (5.72 to 5.83) | 6.77 (6.70 to 6.84) |
| Share of medicine OOP expenditure to total household expenditure (%) | 3.93 (3.87 to 3.98) | 4.10 (4.06 to 4.14) | 4.49 (4.45 to 4.54) |
| Share of health to non-food household expenditure (%) | | | |
| Share of total OOP payments to non-food expenditure (%) | 12.37 (12.20 to 12.55) | 10.82 (10.72 to 10.91) | 11.46 (11.36 to 11.56) |
| Share of medicines OOP payments to non-food expenditure (%) | 10.02 (9.88 to 10.17) | 7.68 (7.62 to 7.75) | 7.60 (7.54 to 7.67) |

Numbers in brackets are 95% CI.
*State-specific and rural–urban-specific consumer price indices were used to convert current prices values at the constant 1999–2000 prices. The current prices values for monthly per capita total OOP are 15.7, 41.8 and 111.2 and for medicine OOP are 12.8, 29.8 and 73.9 (all in INR) for the years 1993–1994, 2004–2005 and 2011–2012, respectively.
OOP, out-of-pocket.

**Table 2** Percentage of households incurring catastrophic payments with respect to total OOP spending and medicines OOP spending, 1993–1994, 2004–2005 and 2011–2012

| Financial health equity measurements | 1993–1994 (%) | 2004–2005 (%) | 2011–2012 (%) | Estimated no of households (2011–2012) |
|---|---|---|---|---|
| Cut-off for catastrophe using total household expenditure | | | | |
| Total OOP Payment >5% | 26.9 (26.6, 27.1) | 28.7 (28.5 to 30.0) | 35.3 (35.0 to 35.6) | 90 107 225 |
| Total OOP Payment >10% | 13.9 (13.8 to14.2) | 14.6 (14.4 to 14.8) | 17.9 (17.7 to 18.2) | 45 691 766 |
| Total OOP Payment >25% | 3.9 (3.8 to 4.0) | 3.5 (3.4 to 3.6) | 4.3 (4.2 to 4.4) | 10 976 234 |
| Medicines OOP Payment >5% | 23.3 (23.0 to 23.5) | 23.4 (23.2 to 23.6) | 27.0 (26.7 to 27.2) | 68 920 540 |
| Medicines OOP Payment >10% | 11.5 (11.3 to 11.7) | 10.2 (10.2 to 10.4) | 11.2 (11.0 to 11.4) | 28 589 261 |
| Medicines OOP Payment >25% | 02.9 (2.8 to 2.9) | 1.6 (1.5 to 1.7) | 1.8 (1.7 to 1.9) | 4 594 703 |
| Cut-off for catastrophe using non-food expenditure | | | | |
| Total OOP Payment >5% | 47.8 (47.5 to 48.1) | 46.5 (46.2 to 46.8) | 53.5 (53.2 to 53.8) | 136 564 775 |
| Total OOP Payment >10% | 34.8 (34.6 to 35.1) | 31.0 (30.7 to 31.2) | 34.9 (34.7 to 35.2) | 89 086 180 |
| Total OOP Payment >25% | 16.7 (16.5 to 16.9) | 11.4 (11.2 to 11.5) | 11.9 (11.7 to 12.1) | 30 376 090 |
| Total OOP Payment >40% | 9.7 (9.5 to 9.9) | 4.7 (4.6 to 4.9) | 4.9 (4.8 to 5.0) | 12 507 802 |
| Medicines OOP Payment >5% | 44.7 (44.4 to 45.0) | 42.5 (42.2 to 42.8) | 46.4 (46.1 to 46.7) | 118 441 225 |
| Medicines OOP Payment >10% | 31.2 (31.0 to 31.5) | 25.5 (25.3 to 25.7) | 26.1 (25.9 to 26.4) | 66 623 189 |
| Medicines OOP Payment >25% | 13.9 (13.7 to 14.1) | 7.1 (7.0 to 7.3) | 6.3 (6.1 to 6.4) | 16 081 459 |
| Medicines OOP Payment >40% | 7.8 (7.6 to 7.9) | 2.2 (2.1 to 2.3) | 1.8 (1.7 to 1.9) | 4 594 703 |

Figures in brackets are 95% CI.
OOP, out of pocket.

consumption expenditure in real terms from INR517 in 1993–1994 to INR794 in 2011–2012. However, during the same period total OOP payments increased by more than 100% from INR26 in 1993–1994 to INR54 in 2011–2012 in real terms. The rise in OOP payments on medicines has been more than 70% during the same period. Consequently, the share of spending on health from households' overall consumption expenditure have registered sharp increase during the past two decades, from a moderate 4.8% during 1993–1994 to nearly 7% in 2011–2012. If we were to net out food expenditure from total household consumption spending, which are considered a necessity, the share of health spending remained stagnant but as high as 11%–12% during the period under consideration. It may be observed that in 2011–2012, medicines alone contributed up to 67% of the total OOP payments.

A higher burden of households' OOP payment is often associated with impoverishment and catastrophe. In table 2, we present a set of catastrophic cut-offs measured as a share of OOP payments to total household consumption expenditure and non-food expenditure. Estimates for both total OOP payments as well as medicine OOP payments by households are presented.

Over one-third of Indian households incurred OOP payments at 5% threshold of total household expenditure in 2011–2012. This percentage was lower in 1993–1994 (27%) and 2004–2005 (28%). At the 25% threshold of total household expenditure, over 4% households reported incurring OOP payments in 2011–2012. This essentially translates to approximately 11 million Indian

households. Out of these, more than 4.4 million households incurred such payments only on account of purchase of medicines. At a lower threshold of 10% of total household expenditure, the number of households facing catastrophe is approximately 46 millions, of which 29 million households incurred catastrophe on account of OOP payments on medicines alone. Considering only non-food expenditure of households as the basic living status variable, approximately similar number of households incurred medicines OOP payments in 2011–2012 with OOP payments being as high as 40% of their non-food expenditure.

Next, we present implications of total and medicine OOP payments on poverty estimates (table 3). To facilitate interpretation, we present three basic head count ratio indicators: (1) gross head count—percentage of population below poverty line, (2) net of OOP head count—percentage of population below poverty line after netting out OOP payments from household consumption expenditure and (3) OOP payments induced poverty which is the difference of the first two reflecting rise in poverty ratio owing to OOP payments. The last two indicators are presented separately for total OOP payments and medicine OOP payments. Table 3 also provides estimates on poverty gap representing extent of poverty deepening in terms of monetary value. All these indicators are estimated using Indian official poverty line (Tendulkar Committee method) and international poverty line of US$1.90 PPP.

The difference in mean head count measure of gross and net poverty ratios reflects the percentage of

**Table 3** Impoverishment indicators due to households' total OOP and medicine spending, India, 1993–1994, 2004–2005 and 2011–2012

| | 1993–1994 | 2004–2005 | 2011–2012 | Estimated population in millions, (2011–12) |
|---|---|---|---|---|
| **Using national poverty line*** | | | | |
| **Head count ratio indicators (%)** | | | | |
| Gross head count | 45.32 (45.03, 45.61) | 37.85 (37.58, 38.12) | 22.17 (21.92, 22.43) | 272 |
| Head count net of total OOP | 49.52 (49.22, 49.81) | 42.68 (42.40, 42.95) | 26.65 (26.38, 26.92) | 327 |
| Total OOP payment induced poverty | 4.20 (4.07, 4.30) | 4.83 (4.71, 4.94) | 4.48 (4.35, 4.60) | 55 |
| Head count net of medicine OOP payment | 48.91 (48.61, 49.20) | 41.54 (41.27, 41.82) | 25.27 (25.00, 25.53) | 310 |
| Medicine OOP payment induced poverty | 3.59 (3.47, 3.69) | 3.69 (3.59, 3.80) | 3.09 (2.99, 3.20) | 38 |
| **Poverty gap indicators (INR current prices)** | | | | |
| Gross poverty gap† | 63.3 (62.9, 63.8) | 103.4 (102.7, 104.2) | 154.2 (152.3, 156.0) | |
| Gap net of total OOP payment‡ | 69.7 (69.3, 70.1) | 115.8 (115.1, 116.5) | 182.8 (181.0, 184.7) | |
| Total OOP payment induced gap‡ | 6.4 (6.3, 6.5) | 12.4 (12.2, 12.6) | 28.6 (28.0, 29.2) | |
| Gap net of medicine OOP payment§ | 68.9 (68.5, 69.3) | 113.7 (113.0, 114.4) | 176.7 (174.9, 178.5) | |
| Medicine OOP payment induced gap§ | 5.6 (5.5, 5.7) | 10.3 (10.1, 10.4) | 22.5 (22.0, 23.0) | |
| **Using international poverty line¶** | | | | |
| **Head count ratio indicators (%)** | | | | |
| Gross head count | 40.96 (40.67 to 41.24) | 33.07 (32.81 to 33.34) | 18.37 (18.13, 18.61) | 225 |
| Head count net of total OOP payment | 44.92 (44.63, 45.21) | 37.38 (37.11, 37.65) | 22.41 (22.16, 22.67) | 275 |
| Total OOP payment induced poverty | 3.97 (3.85, 4.08) | 4.31 (4.19, 4.42) | 4.04 (3.92, 4.16) | 50 |
| Head count net of medicine OOP payment | 44.35 (44.06, 44.64) | 36.34 (36.08, 36.61) | 21.37 (21.11, 21.62) | 262 |
| Medicine OOP payment induced poverty | 3.39 (3.29, 3.50) | 3.27 (3.17, 3.68) | 2.99 (2.89, 3.10) | 37 |
| **Poverty gap indicators (INR current prices)** | | | | |
| Gross poverty gap† | 59.3 (58.9, 59.7) | 96.1 (95.3, 96.8) | 150.7 (148.8, 152.7) | |
| Gap net of total OOP payment‡ | 65.4 (64.9, 65.8) | 107.5 (106.8, 108.3) | 177.0 (175.1, 179.1) | |
| Total OOP payment induced gap‡ | 6.1 (6.0, 6.2) | 11.5 (11.2, 11.7) | 26.3 (25.7, 27.0) | |
| Gap net of medicine OOP payment§ | 64.6 (64.2, 65.1) | 105.8 (105.0, 106.5) | 172.0 (170.0, 174.0) | |
| Medicine OOP payment induced gap§ | 5.3 (5.2, 5.4) | 9.7 (9.5, 9.9) | 21.3 (20.7, 21.8) | |

Figures in  brackets are  95 %  CI .
*Based on Tendulkar Committee methods (poverty lines range between INR695 in Odisha and INR1018 in Kerala in rural and INR861 in Odisha and INR1169 in Haryana in urban areas).
†Only for poor.
‡Only for poor net of total OOP.
§Only for poor net of medicine OOP.
¶Using US$1.90 PPP at 2011–2012 prices and mixed recall period of household consumption expenditure (INR equivalent to US$1.90 PPP are 771.21 in rural and 945.41 in urban areas).
OOP, out of pocket; PPP, purchasing power parity.

population falling below poverty line because of households' OOP payments on healthcare. The head count ratio of households impoverished due to OOP payments was 3.97% during 1993–1994, which inched up to 4.30% in 2004–2005 while in 2011–2012 it was at 4.04%, as per international poverty line. In terms of Indian state-specific official poverty lines, percentage of households falling below poverty line increased from 4.19% in 1993–1994 to 4.48% in 2011–2012. This translates to 55 million persons in 2011–2012. Out of this, approximately 38 million became poor only because they had to purchase medicines through OOP payments. Using

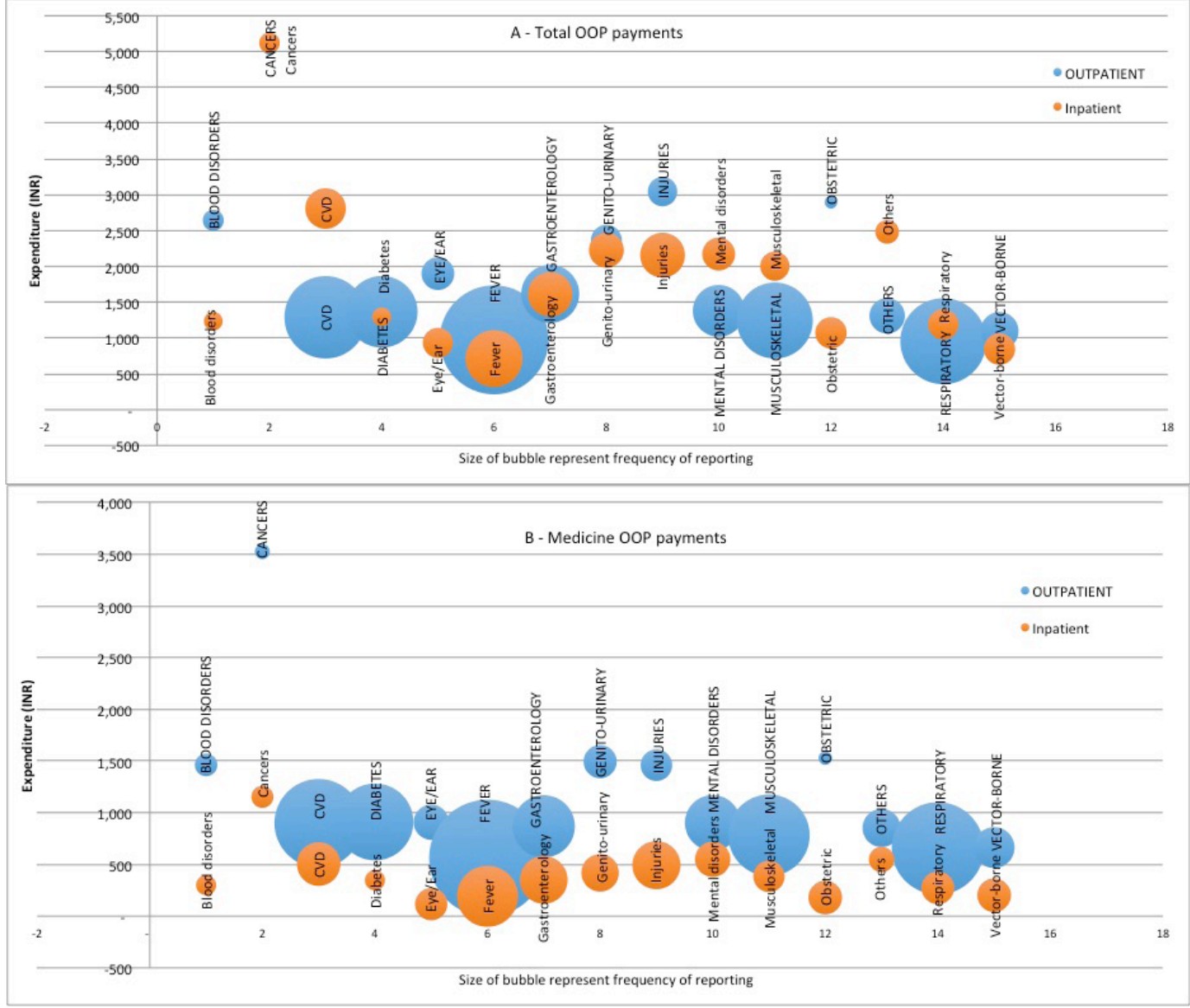

**Figure 1** Frequency and monthly per person (A) total OOP and (B) medicine OOP spending on select disease conditions, 2014. CVD, cardiovascular disease.

the same measurement, the head count measure for households OOP payments on medicines appear to have marginally declined from 3.59% in 1993–1994 to 3.09% during 2011–2012 using the international poverty line. As far as poverty gap is concerned, based on the Indian official poverty line, total OOP payments and OOP payments on medicines resulted in poverty deepening among poor by INR29 and INR23, respectively, in 2011–2012. Further poverty deepening because of total and medicines OOP payments sharply increased in 2012 compared with that in the years 2004–2005 and 1993–1994.

### OOP expenditure by disease conditions

We also conducted a disaggregated analysis on disease-wise expenditure with reference to total OOP payments and medicines OOP payments and by type of care—inpatient care versus outpatient care. The survey results suggested that most common health condition for

seeking outpatient care was fever (22.7%) and for inpatient care was childbirth (27.3%). In addition, our estimates suggest that households incurred highest monthly per capita OOP spending both for inpatient and outpatient care on account of cancer treatment (INR5054 and INR5121, respectively) followed by injuries for outpatient care (INR3045) and cardiovascular events for inpatient care (INR2808).

We also mapped disease-wise expenditure, frequency of healthcare utilisation and type of care to demonstrate that hospitalisation and outpatient care can lead to catastrophe and impoverishment of households (figure 1). For example, our estimates suggest that monthly per capita medicines OOP payments for cancer care were significantly higher in outpatient care as compared with the inpatient care. However, as far as total OOP spending for cancer treatment is concerned,

it is almost similar across inpatient and outpatient but significantly higher compared with that for other disease conditions. In contrast, in respect to cardiovascular conditions, medicines OOP payments were similar for both inpatient and outpatient treatment, but total OOP payments were significantly higher for inpatient treatment as against outpatient treatment. In treatments involving gastroenterology conditions, however, both medicines OOP payments and total OOP payments were higher for outpatient compared with inpatient treatment. Similarly, for mental disorders, medicines OOP payments were higher for outpatient care compared with inpatient, but total OOP payments were almost similar both for outpatient and inpatient treatment. Therefore, it is noted that the average monthly medicines OOP payments were consistently higher for outpatient care as compared with inpatient care among key disease conditions. A relatively higher frequency of outpatient treatment visits compared with inpatient treatment coupled with a significantly larger medicines OOP payment may yield a higher incidence of catastrophe. A detailed estimate of prevalence and OOP payments by disease conditions cross-classified by inpatient and outpatient care are presented in (online supplementary tables A–II).

Further, we plotted outpatient and inpatient OOP payments and medicine-related OOP payments with respect to households 'usual' consumption expenditure. In figure 2, households are ranked from the poorest to the richest on the X-axis based on their mean monthly per person consumption expenditure and on the Y-axis mean monthly per person OOP expenditure (total and medicine) are measured separately for outpatient and inpatient. It is observed that for a number of households, average monthly outpatient expenditure is significantly higher in relation to household's non-medical consumption expenditure and the frequency of such events is higher in outpatient care as compared with inpatient care. In figure 2, the concentration of red (total OOP payment) and green (medicine OOP payment) spikes above the consumption expenditure on the right-hand side of the graph reflects that even among richer households total OOP and medicine OOP payments are significantly higher than total non-medical consumption expenditure of households. Moreover, concentration of medicine OOP payments above households' non-medical consumption expenditure is more prominent in case of outpatient compare to the inpatient episodes.

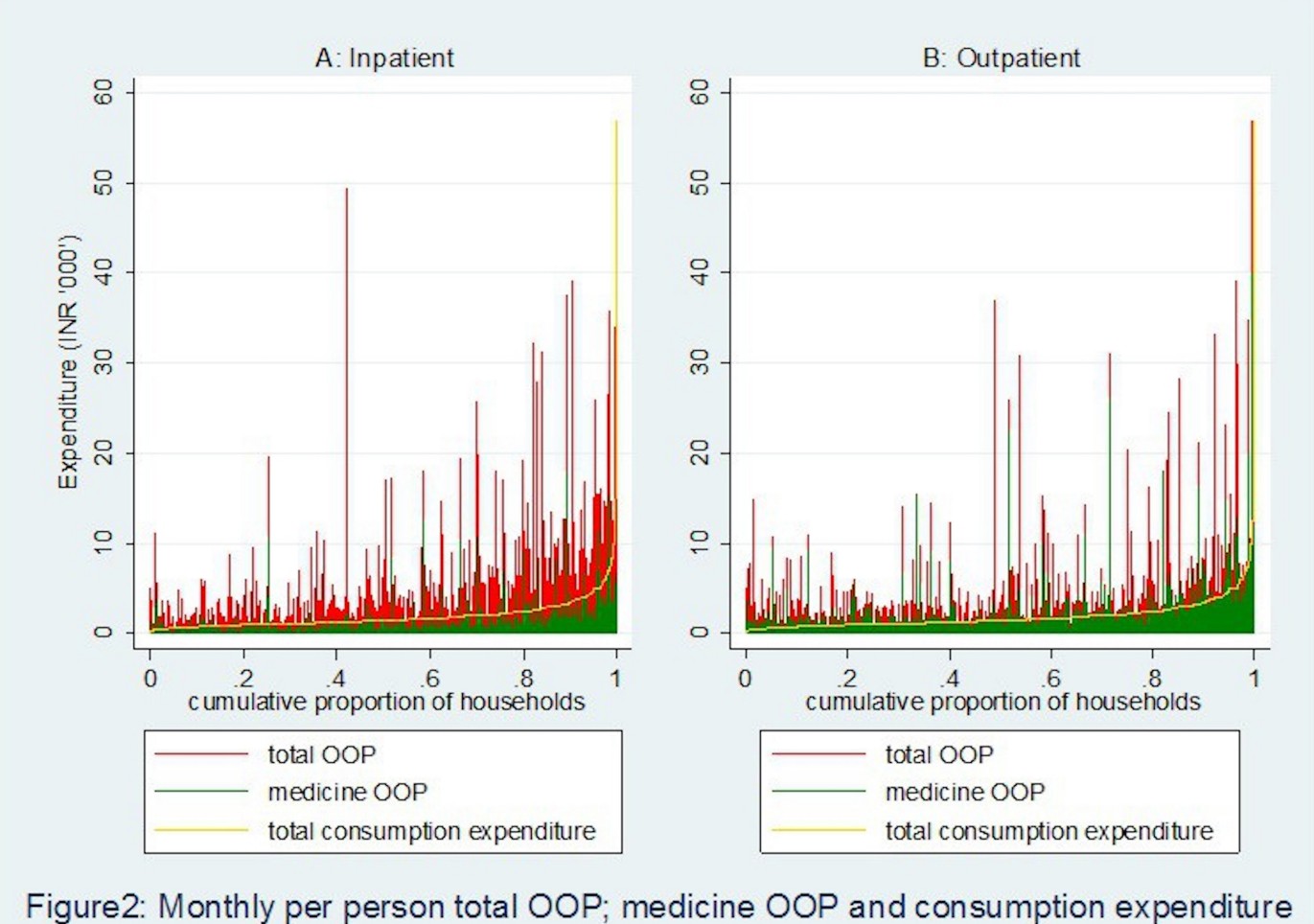

Figure 2    Monthly per person total out-of-pocket (OOP) payment, medicine OOP payment and consumption expenditure.

## DISCUSSION AND CONCLUSION

Using standard methods of measuring catastrophe and impoverishment,[17 18] this paper demonstrates the financial burden of households' OOP payments on medicines in India, spanning two decades from 1993–1994 to 2011–2012. To our knowledge, this is a first attempt to link medicines' OOP spending to key diseases conditions. Two trends stand out clearly from our findings. First, the households' impoverishment on account of OOP expenditure is rather high and continued to be so during the last two decades. The impoverishment burden is largely driven by households spending on medicines, which accounted for over three-fourth of all medical impoverishment in India. One of the reasons could be, compared with OOP payments on medicines, hospitalisation/bed charges in India are comparatively low, and often subsidised in the public sector. Second, as far the catastrophe measurement is concerned, applying a 10% threshold of OOP payment on overall consumption expenditure, an estimated 18% of Indian households appear to suffer financial catastrophe. Medicines' OOP expenditure alone contributed to an estimated 11% of financial catastrophe. In absolute numbers, this translates to a scenario where an estimated 46 million households appear to face catastrophic expenditure on account of OOP payments while 29 million households faced such hardship because they had to pay for medicines from their pockets.

Recent evidence from the National Health Accounts for India points out that during 2013–2014, an estimated INR1331 per capita was spent on medicines, while households alone contributed INR1200 per capita, accounting for 90% of all medicines expenditure in the country.[1] On the other hand, past evidence about government expenditure on medicines in India, underscores that, on an average government spent about 10% of health expenditure on medicines. However, the national average masks significant underspending on medicines by several state governments, with many reportedly spending less than 5% of their health budgets.[6] Besides poor allocation of resources, except for a couple of Indian states, drug procurement and supply chain system is inefficient and ineffective leading to acute shortages of key essential medicines and chronic stock-outs in public health facilities.[21–24] This situation has resulted in physical unavailability of medicines. Drawing evidence from large sample surveys for the period from 1986–1987 to 2004, it is reported the physical barrier to access to key essential medicines worsened during this period.[15] Supply of free drugs in government health system in the outpatient care setting declined sharply from about 18% in 1986–1987 to 5% in 2004. For the same period, drugs prescribed during hospitalisation for free also declined significantly from one-third to about 9%.[6] As a result, it is pointed out the number of hospitalisation episodes in which an ailing population paid OOP payment, has risen dramatically from about 41% to close to 72%.[6] Further, it was observed that from the period spanning mid-1990s to 2004, patients visiting government health facilities did not receive medicines in over one-fourth of outpatient episodes. Affordability of medicines is an important access indicator, because it translates into poor access or no access for people who have low purchasing power.[3] The consumer behavioural theory also predicts that raising the price (via high OOP expenditure on medicines or high copayment) for a service in the public health sector will move more consumers into the private sector, depending on the elasticity of substitution and transaction costs in the public sector.[25]

In view of inadequate availability of medicines in government health facilities, households end up accessing private facilities where they end up incurring significant OOP payment, in the absence of any financial risk protection. Past evidence suggests that the trend has sharpened in the last couple of decades. For instance, the percentage of population accessing private facilities for inpatient and outpatient treatment has accelerated significantly between 1986–1987 and 2004. It may be observed that households accessing private hospital for inpatient care increased from around 40% to nearly 60% in rural India while urban India reported a rise from 40% to 68%.[26] During the same period, outpatient care visits in private facilities remained high at around 75% in 1986–1987 in rural India and 73% in urban India stepped up to 78% and 80%, respectively, for rural and urban India.[3]

The other critical evidence emerging from this paper focuses on disease-specific medicines expenditure. The results demonstrate a pattern where households' medicine spending is concentrated on low frequency, high-value spending and high frequency, high-value spending. By disease-wise classification, expenditure on treatment of cancers, CVDs and injuries, both for outpatient and inpatient care dominate the spending pattern. Available literature confirms such an expenditure pattern, wherein the share of NCDs (CVD, diabetes, cancer, mental illness, injuries and others) in OOP health expenses has increased from 31.6% in 1995–1996 to 47.3% in 2004.[12] The literature further indicates high odds of catastrophic hospitalisation expenditures for certain NCDs. For example, the odds for catastrophic expenditure in cancer are nearly 170% greater, for CVDs and injuries nearly 22% greater than the odds due to infectious diseases. Other studies on CVDs highlighted that CVD-affected households had more outpatient visits and inpatient stays, spent extra money per hospitalisation[11] and have high probability of incurring catastrophic expenditure compared with those using inpatient facilities for communicable conditions.[27] Another Indian study on socioeconomic inequalities in financing of diabetes and CVD reported that OOP payments for hospital treatment claimed a large share of annual household expenditures; 30% for CVD and 17% for diabetes.[28] In respect to injuries (both road traffic and non-road traffic), high incidence of catastrophic expenditure was 30%, and was significantly higher among those belonging to the lowest income quartile and with an inpatient stay greater than 7 days.[29]

Although public facilities have slightly stepped up their share in outpatient care in recent years, private sectors continue to dominate both in outpatient and inpatient care in India.[30] As an increasing share of households' access private health facilities, private retail pharmacies have become a major source of supply of key essential medicines. While availability of medicines may per se is not a challenge in the private healthcare setting, affordability appears to act as critical barrier.[31] Thus, pricing of medicines and regulation around retail medicine prices becomes a key factor in improving affordability and thereby leading to a reduction in medicine-related OOP payment burden. Although India had a progressive retail price cap policies since 1979, but over the years a policy of deregulation was followed.[32] In 2013, the government of India promulgated the Drugs Price Control Order (DPCO), 2013 (DPCO, 2013) which primarily brought all essential drugs, based on National List of Essential Medicines, 2011, under price control.[33] An evaluation of new price regulation has highlighted that, while few of the medicines (37) had an increase in sales volume attributable to DPCO, majority of the medicines (52) had a negative impact on their sales volume due to DPCO. Overall, the DPCO may have had a negative impact in terms of sales volume of medicines under price control.[34] Given that the sales volume of price-capped medicines has declined, households OOP spending may continue to increase since over 80% of retail pharmacy market is not price capped.

In order to improve access to healthcare and to provide financial risk protection to households, the central government and several state governments have been implementing a publicly funded health insurance programmes since 2007, whose primary aim was to provide cashless treatment to economically vulnerable households for hospitalisation episodes. Emerging evidence from micro-level as well as macro-level studies point to a trend where such insurance schemes appear to have improved access to hospital care but have been ineffective in preventing financial catastrophe and impoverishment to households.[2 35 36] These bodies of evidence are in line with our findings that hospitalisation-based treatment cost constitutes only one-third of India's morbidity burden. Despite implementation of several health insurance schemes, a majority of Indian population continues to incur a relatively significant medicines OOP payment while seeking outpatient care. It would be pertinent to highlight that the frequency of hospitalisation is considerably smaller than outpatient visits in general, especially for NCDs, which are chronic in nature that require multiple consultations and long-term or lifelong medication support. Such medical conditions result in catastrophic expenditure for households even in the absence of hospitalisation episodes. Moreover, since a relatively larger proportion of population seeks outpatient care in private facilities, which is often multiple times expensive than public health facilities, we observe a disproportionately higher burden of medicine-related OOP payment for outpatient care.

The evidence presented in this paper, however, suffers from a few limitations. The first set of challenge relates to comorbidities and associated expenditure. In respect to inpatient cases, since NSSO data capture disease expenditure separately for various disease conditions, the issue of comorbid conditions did not play major role. However, for outpatient cases, we had to adopt apportioning technique to handle comorbid conditions. The second set of challenge pertains to the potential recall bias for disease-specific expenditures, which cannot be ruled out especially for hospitalisation treatment since the recall period is a longer time span of 365 days. Lastly, although there are significant state-level and rural–urban differentials in the estimates presented in this paper, we focused on the all India average and believe that the state-level and rural–urban analyses could be a potential research for the future.

The foregoing underlines several policy interventions and programme design that were conceived and implemented in the recent past to provide financial risk protection to households. However, gross underinvestment in the public health system in past had led to inadequate prepayment and risk pooling measures.[26] Several policy interventions and programme redesign are required to reverse the trend of high OOP expenditure for healthcare in India. An efficient and reliable medicines supply chain model existed for over two and half decades in the state of Tamil Nadu which was replicated in the state of Rajasthan in 2012 and has been instrumental in improving access to medicines in the frontline facilities in these two states.[37] Such policies and programmes governing public health facilities are critical. The National Health Policy 2017 also highlighted the need for providing free medicines in public health facilities by stepping up funding and improving drug procurement and supply chain mechanisms.[38] A recent pronouncement by the government intends to bring legislation for physicians to prescribe drugs only in generic names, also holds promise for reducing households' OOP payments on medicines and ultimately providing financial risk protection. To sum up, both national and state governments' intervention is required for providing free medicines in public health facilities along with expanding the mechanism of price capping of key essential medicines in the private market.

**Contributors** AK and SS conceived the idea, designed the analysis, conducted data analysis and wrote the first draft of the paper. AK, SS and HHF conducted the literature review and the interpretation of the results. AK, SS and HHF revised and edited the manuscript to its final stages. All the authors approved the final manuscript version.

**Funding** Habib Hasan Farooqui is supported by a Welcome Trust Capacity Strengthening Strategic Award to the Public Health Foundation of India and a consortium of UK universities.

**Competing interests** None declared.

**Patient consent** Not required.

**Provenance and peer review** Not commissioned; externally peer reviewed.

**Data sharing statement** The data used for the analysis are available in public domain.

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
