## [Reviewer comments · BMJ Open]

ARTICLE DETAILS

TITLE (PROVISIONAL)	Financial burden of medicine out of pocket payments: Implications for living status of Indian households
AUTHORS	Selvaraj, Sakthivel Farooqui, Habib Karan, Anup

VERSION 1 – REVIEW

REVIEWER	Jahangir Khan Liverpool School of Tropical Medicine United Kingdom
REVIEW RETURNED	07-Aug-2017

GENERAL COMMENTS	The paper is informative considering volume of information, but not articulated adequately for addressing relevant policy issues. While outcomes have been estimated, explanations behind such outcomes are not clearly stated. Following comments should be considered in the next version: 1. The authors could not successfully develop the rationale for studying OOP health expenditure for medicines. Medicines are complementary to health care along with doctor's consultations, diagnostic tests, travels etc. The authors should have focused on the reasons for high costs of medicines while developing the rationale.2. Methodology mainly focused on data used. The authors should put emphasis on how variables were defined and constructed from the datasets.3. Disease-specific analysis has an added value. But it is not clear how co-morbidity was considered in the analysis. Number of disease-cases should be stated in the table.4. In table 1 authors presented values in current and constant prices. It is confusing and they should choose an appropriate one, preferably in constant price.5. Figure 1 can be confusing for readers. It looks like OOPs are on top of total consumption expenditure. Authors can look into figure 2 in the following article, which shows how household expenditure drops when OOPs are excluded: Khan JAM, Ahmed S, Evans TG. 2017. Catastrophic healthcare expenditure and poverty related to out-of-pocket payments for healthcare in Bangladesh—an estimation of financial risk protection of universal health coverage, Health Policy and Planning, 1–9 doi:
--

	10.1093/heapol/czx048. 6. Data in different points of time might be collected differently or the variables were defined differently. Authors should have discussed if data across the periods are comparable and useful for a trend analysis. The limitations in this regard should have been discussed. 7. Authors can find more references in the relevant study area by checking the article above (Khan et al. 2017) 8. Discussion should be better structured. Results of the study, explanations, limitations, significance of OOP for medicine (while other authors studied OOPs in total) should be clearly discussed for acceptance. I am reluctant and find no strong reason in this paper why OOPs for medicines has been studied.
--	---

REVIEWER	Kiran Raj Pandey Center for Health and Social Sciences University of Chicago USA
REVIEW RETURNED	13-Sep-2017

GENERAL COMMENTS	1. The authors present timely work given the current focus on ensuring universal health coverage and reducing financial burden due to health expenditures in low and middle income countries . The authors' focus on comparing out of pocket payments in the outpatient and inpatient settings is useful as is the comparison of financial burden due to several disease conditions. Also, a comparative analysis of financial burden due to health expenditure over two decades is very helpful, especially given that measures of financial burden tend to be arbitrary and have lesser utility when used as stand alone point estimates rather than time trends. However, several points need attention. 2. Paper needs careful copy editing. The manuscript appears to have been prepared hastily and has several obvious copy editing errors including misspellings. Example: on several occasions, the authors simply write "out of pocket" or OOP when they mean "OOP payment". Several other examples abound. 3. The authors' claim that "This study is the first one to produce evidence on catastrophic and poverty impact of out-of-pocket payments for medicine" is not true even in the Indian context, let alone the global context. 4. The authors claim that "The study also highlights for the first time that out-of-pocket payments on medicines in outpatient care is far greater burden for households than expenditure on hospitalization care" is not entirely true. This has been established before in the Indian context, even in work that the authors cite. 5. Citations have been placed before punctuation marks. Needs to be the other way around. 6. Table 1. Table headings (those in bold should be above the right 3 columns instead of over the left column), unless that is not what the authors meant, which is also problematic. More generous foot
---

notes explaining the tables would be helpful.

7. The figures on Table 2 are percentages and need to be labeled as such either on the column header or on the footnote. It is helpful to mention in the footnote that the currency that is being implied (INR) and the year that it was inflated to.

8. Table 2 headings "Threshold using on ..." are confusing. Could be worded better.

9. It is unclear if separate poverty lines were used for rural and urban India. The Tendulkar committee had different poverty thresholds for the two in each state.

10. Furthermore, given that both the head count ratio impoverishment method as well as catastrophic expenditure methods aren't able to elicit depth of poverty, the use of poverty gaps might have been beneficial. This is easy to calculate with the micro-data that the authors had access to, using methods outlined by Wagstaff and others. The manuscript would benefit if this were at least mentioned in the discussion section.

11. Its better to not use terms like "Cardio" and "Gastro" as these terms are rather colloquial and are not appropriate in an academic paper. They are also likely to confuse readers, especially since they aren't really used in clinical settings outside of India.

12. A fact that is hidden in these comparisons of household expenditures on outpatient and inpatient healthcare expenditures is that in India, inpatient stay in public hospitals tends to be subsidized by the government (the amount that is charged is hardly reflective of even the marginal cost the government has to bear to produce an inpatient stay) while expenditures on medications are more likely to be not subsidized, especially in the outpatient settings. This is one of the major reasons why patients appear to have way higher medication related or outpatient visit related financial burden than inpatient visit related financial burden. It is helpful to mention this caveat in the discussion for the benefit of readers that are in a setting that might have a different pattern of costs on inpatient and outpatient stays. In other settings like the United States for example, where inpatient stays can cost up to thousands of dollars per night, as expensive as outpatient visits and medications may be, they are not likely to be as financially burdensome as inpatient stays.

13. Outpatient visits may be more burdensome than inpatient visits because more outpatient visits are in the private sector (around 75% as the authors point out; and also more expensive), than inpatient visits (around 40% to 60%). Visits to the private sector are mostly out of pocket unless covered by insurance. Need to mention this for the benefit of readers outside of the Indian context, or else they are likely to draw wrong conclusions.

14. Also hospital care (and thus inpatient costs) tends to be episodic while outpatient medication costs in the case of chronic diseases are recurring and long term. Between the early 1990s and the 2010s, according to the Global Burden of Diseases estimates for India, there has been a clear increase in the burden of chronic diseases. These diseases tend to have high medication and outpatient related expenses while acute illnesses like lower respiratory tract infections are likely to lead to a short inpatient stay and a short course of

	medications. These changing epidemiological trends are clearly likely to have had some effect in the comparative time trends of inpatient versus outpatient costs. 15. Another caveat to mention in the discussion is that all these OOP expenditures are reported and are likely to suffer from recall bias and are especially likely to be inaccurate when attributed to specific disease conditions or anatomical systems.
--	---

REVIEWER	SANJAY K MOHANTY INTERNATIONAL INSTITUTE FOR POPULATION SCIENCES, MUMBAI
REVIEW RETURNED	17-Sep-2017

GENERAL COMMENTS	Research issue addressed in the paper is relevant. However, review of earlier studies and limitation of consumption survey and the ratio method in defining the CHS need to be elaborated. Following are my comments  1. Household health expenditure collected as a part of consumption expenditure are likely to underestimate the health spending of the households. 2. There is no variable on reimbursement of health expenditure in consumption schedule. Hence it is difficult to say whether the expd reported by household is OOPE or total household expenditure 3. The definition of CHS used in the paper is not appropriate. Because, we know the limitation of such method in underestimating the CHS of poor. Any expd on health to those below poverty line is catastrophic. 4. State differentials or by characteristics of households are not analysed 5. Need to mention whether the descriptive and CHS is based on total sample or on those who spend on health. A significant hhd did not spend any money on health care. 6. Catastrophic Payments and Impoverishment due to Out-of-Pocket Health Spending by Soumita Ghosh appeared in EPW 2011 dd for 1993-94 and 2004. This need to be different from beyond expending by 2011-12. NOVEMBER 19, 2011 vol xlvii no 47 , EPW 7. The word total OOPS is confusing. Should it not be used as average household health spending 8. The same way , should it not be used as MPCE at current prices,
---

VERSION 1 – AUTHOR RESPONSE

Dear Sir,

We would like to thank editor and reviewers for their constructive comments. We have revised the manuscript in the light of the valuable comments and suggestions made by the reviewers. Please find below point wise clarification and details of the modification made in the manuscript as per the reviewers comments.

Regards,
Thank you,
Habib Farooqui

Editorial requests:

1. Please revise your title to indicate the research question, study design, and setting. This is the preferred format of the journal.

Response: The title has been modified on suggested lines.

2. Please ensure that you improve the quality of language in your manuscript, either with the assistance of an English-speaking colleague or with a professional copyediting agency.

Response: The language has been edited appropriately for syntax and grammar.

Reviewer(s)' Comments to Author:

Reviewer: 1

Reviewer Name: Jahangir Khan

Institution and Country: Liverpool School of Tropical Medicine, United Kingdom

Please state any competing interests or state 'None declared': None declared.

Please leave your comments for the authors below

The paper is informative considering volume of information, but not articulated adequately for addressing relevant policy issues. While outcomes have been estimated, explanations behind such outcomes are not clearly stated. Following comments should be considered in the next version:

1. The authors could not successfully develop the rationale for studying OOP health expenditure for medicines. Medicines are complementary to health care along with doctor's consultations, diagnostic tests, travels etc. The authors should have focused on the reasons for high costs of medicines while developing the rationale.

Response: Thank you for the suggestion. You would appreciate that in India medicine expenditure constitutes upto 80% of the total healthcare cost in outpatient setting. Other components like consultation and diagnostics are miniscule, hence the focus of the paper is on the outpatient medicines OOP payments. In addition, the poor and vulnerable section relies on OOP for making healthcare payments. Since, the income level and living status of the poor population in India is already low, in absence of financial risk protection, the OOP payments results into impoverishment of households even in absence of high cost of medicines. The focus of the paper is hence on relative contribution of expenditure on medicines across various disease conditions and its impact on households. We have improved the rationale for study of medicines in the introduction section accordingly (Page 4 of revised manuscript)

2. Methodology mainly focused on data used. The authors should put emphasis on how variables were defined and constructed from the datasets.

Response: Detailed methodology is provided in supplementary material section. However, we have also included method of constructing of variables and outcome indicators in the manuscript. (Page 6 of revised manuscript)

3. Disease-specific analysis has an added value. But it is not clear how co-morbidity was considered in the analysis. Number of disease-cases should be stated in the table.

Response: Thank you for your comments. In NSSO data, especially for inpatient cases, the disease expenditure is captured separately for various disease conditions even in cases of co-morbidity, whereas for outpatients we adopted apportioning technique. We believe this has partially addressed the reviewers concerns. Still, we have added this as limitation of the study. (Page 16 of revised manuscript)

4. In table 1 authors presented values in current and constant prices. It is confusing and they should choose an appropriate one, preferably in constant price.

Response: We have removed current prices data in the revised draft. (Table 1 of revised manuscript)

5. Figure 1 can be confusing for readers. It looks like OOPs are on top of total consumption expenditure. Authors can look into figure 2 in the following article, which shows how household expenditure drops when OOPs are excluded:

Khan JAM, Ahmed S, Evans TG. 2017. Catastrophic healthcare expenditure and poverty related to out-of-pocket payments for healthcare in Bangladesh—an estimation of financial risk protection of universal health coverage, *Health Policy and Planning*, 1–9 doi: 10.1093/heapol/czx048.

Response: Usually the OOP expenditure is part of consumer expenditure; this may also be the case in the article cited by the reviewer. In India, Consumer Expenditure Survey (CES) by the NSSO too collects healthcare expenditure as part of total consumption expenditure. However, in the health survey by the NSSO 2014, the data for OOP payments and consumer expenditure are separately recorded and OOP payment is sometimes even higher than total consumption expenditure (termed as usual monthly per capita household expenditure), which is reflected in the figure 1. Hence, we plotted the data as it is, on account of lack of information, whether the OOP expenditure is part of consumption expenditure or not. In fact we wanted to highlight this nuance in the figure 1. (Page 13 of revised manuscript)

6. Data in different points of time might be collected differently or the variables were defined differently. Authors should have discussed if data across the periods are comparable and useful for a trend analysis. The limitations in this regard should have been discussed.

Response: We have used different points of time data only for CES. CES follows exactly same definitions and sampling design across different survey rounds. This has been clarified now in the methods section. For the health survey, we have used only round of data, hence the question of comparability should not be a concern. (Page 5 of revised manuscript)

7. Authors can find more references in the relevant study area by checking the article above (Khan et al. 2017)

8. Discussion should be better structured. Results of the study, explanations, limitations, significance of OOP for medicine (while other authors studied OOPs in total) should be clearly discussed for acceptance. I am reluctant and find no strong reason in this paper why OOPs for medicines has been studied.

Response: Thank you for the valuable comments. We have updated the discussion in light your comment. We have revised manuscript highlighting this as contribution to existing literature on medicine related impoverishment with time trend analysis and additional dimension of disease specific analysis.

We are highlighting that medicine share is single most important driver of healthcare cost and has poverty impact.

Reviewer: 2

Reviewer Name: Kiran Raj Pandey

Institution and Country: Center for Health and Social Sciences, University of Chicago, USA

Please state any competing interests or state 'None declared': None

Please leave your comments for the authors below

1. The authors present timely work given the current focus on ensuring universal health coverage and reducing financial burden due to health expenditures in low and middle income countries . The authors' focus on comparing out of pocket payments in the outpatient and inpatient settings is useful as is the comparison of financial burden due to several disease conditions. Also, a comparative analysis of financial burden due to health expenditure over two decades is very helpful, especially given that measures of financial burden tend to be arbitrary and have lesser utility when used as stand alone point estimates rather than time trends.

Response: Thanks for recognizing the importance of the study and useful comments.

However, several points need attention.

2. Paper needs careful copyediting. The manuscript appears to have been prepared hastily and has several obvious copy editing errors including misspellings. Example: on several occasions, the authors simply write "out of pocket" or OOP when they mean "OOP payment". Several other examples abound.

Response: We have revised the manuscript in the light of the comments and have replaced OOP with OOP payment or OOP expenditures.

3. The authors' claim that "This study is the first one to produce evidence on catastrophic and poverty impact of out-of-pocket payments for medicine" is not true even in the Indian context, let alone the global context.

Response: This has been revised. We have revised manuscript highlighting this as contribution to existing literature on medicine related impoverishment with time trend analysis and additional dimension of disease specific analysis. (Page 7 of revised manuscript)

4. The authors claim that "The study also highlights for the first time that out-of-pocket payments on medicines in outpatient care is far greater burden for households than expenditure on hospitalization care" is not entirely true. This has been established before in the Indian context, even in work that the authors cite.

Response: This has been revised. We have revised the content of the manuscript highlighting the dimension of time trend analysis of healthcare expenditure on inpatient and outpatient care. (Page 7 of revised manuscript)

5. Citations have been placed before punctuation marks. Needs to be the other way around.

Response: Done.

6. Table 1. Table headings (those in bold should be above the right 3 columns instead of over the left column), unless that is not what the authors meant, which are also problematic. More generous foot notes explaining the tables would be helpful.

Response: This is not the heading of the columns; it is heading for the first to items in the row. We have added % sign at appropriate places in the table for clarity. (Table 1 of revised manuscript)

7. The figures on Table 2 are percentages and need to be labeled as such either on the column header or on the footnote. It is helpful to mention in the footnote that the currency that is being implied (INR) and the year that it was inflated to.

Response: We have added % in the column heading in the table. We don't think INR needs to be mentioned as the outcomes are reported as proportions.

8. Table 2 headings "Threshold using on ..." are confusing. Could be worded better.

Response: We have changed the term from 'threshold' to 'cutoff' for clarity in the table. (Table 2 of revised manuscript)

9. It is unclear if separate poverty lines were used for rural and urban India. The Tendulkar committee had different poverty thresholds for the two in each state.

Response: We have used state-specific rural and urban poverty line separately for our analysis. This has been clarified now in the methods section. (Page 6 of revised manuscript)

10. Furthermore, given that both the head count ratio impoverishment methods as well as catastrophic expenditure methods aren't able to elicit depth of poverty, the use of poverty gaps might have been beneficial. This is easy to calculate with the micro-data that the authors had access to, using methods outlined by Wagstaff and others. The manuscript would benefit if this were at least mentioned in the discussion section.

Response: Thank you for the comment. We have now included poverty gap indicator in the table 3 and have discussed the results accordingly. (Table 3 of revised manuscript)

11. Its better to not use terms like "Cardio" and "Gastro" as these terms are rather colloquial and are not appropriate in an academic paper. They are also likely to confuse readers, especially since they aren't really used in clinical settings outside of India.

Response: We have revised the terminology in the manuscript appropriately.

12. A fact that is hidden in these comparisons of household expenditures on outpatient and inpatient healthcare expenditures is that in India, inpatient stay in public hospitals tends to be subsidized by the government (the amount that is charged is hardly reflective of even the marginal cost the government has to bear to produce an inpatient stay) while expenditures on medications are more likely to be not subsidized, especially in the outpatient settings. This is one of the major reasons why patients appear to have way higher medication related or outpatient visit related financial burden than inpatient visit related financial burden. It is helpful to mention this caveat in the discussion for the benefit of readers that are in a setting that might have a different pattern of costs on inpatient and outpatient stays. In other settings like the United States for example, where inpatient stays can cost up to thousands of dollars per night, as expensive as outpatient visits and medications may be, they are not likely to be as financially burdensome as inpatient stays.

Response: Thank you for the valuable comments. We have updated the discussion in light of your comment. We would like to highlight that we are not comparing the expenditure on medicines with other components of the healthcare cost. We are highlighting that medicine share is single most important driver of healthcare cost and has poverty impact.

13. Outpatient visits may be more burdensome than inpatient visits because more outpatient visits are in the private sector (around 75% as the authors point out; and also more expensive), than inpatient visits (around 40% to 60%). Visits to the private sector are mostly out of pocket unless covered by insurance. Need to mention this for the benefit of readers outside of the Indian context, or else they are likely to draw wrong conclusions.

Response: Thank you for the comment. We have included this as part of our discussion. (Page 14 of revised manuscript)

14. Also hospital care (and thus inpatient costs) tends to be episodic while outpatient medication costs in the case of chronic diseases are recurring and long term. Between the early 1990s and the 2010s, according to the Global Burden of Diseases estimates for India, there has been a clear increase in the burden of chronic diseases. These diseases tend to have high medication and outpatient related expenses while acute illnesses like lower respiratory tract infections are likely to lead to a short inpatient stay and a short course of medications. These changing epidemiological trends are clearly likely to have had some effect in the comparative time trends of inpatient versus outpatient costs.

Response: Thank you for the comment. We have included this as part of our discussion. (Page 15 of revised manuscript)

15. Another caveat to mention in the discussion is that all these OOP expenditures are reported and are likely to suffer from recall bias and are especially likely to be inaccurate when attributed to specific disease conditions or anatomical systems.

Response: Thank you for the comment. We have included this as part of our limitation for disease specific estimates in the manuscript.

Reviewer: 3

Reviewer Name: Sanjay K Mohanty

Institution and Country: INTERNATIONAL INSTITUTE FOR POPULATION SCIENCES, MUMBAI

Please state any competing interests or state 'None declared': NONE

Please leave your comments for the authors below

Research issue addressed in the paper is relevant. However, review of earlier studies and limitation of consumption survey and the ratio method in defining the CHS need to be elaborated. Following are my comments

1. Household health expenditure collected as a part of consumption expenditure are likely to underestimate the health spending of the households.

Response: You are right that OOP is a part of CES, but it is difficult to comment whether the OOP is underestimated in CES or overestimated in health survey.. However, the NSSO report itself mentions that total household expenditure is underestimated in health survey. Hence, it is not appropriate to calculate poverty ratio and catastrophe from health survey. This is main reason we used CES, which is available upto 2011-2012 for poverty and catastrophe estimates. (Page 5-6 of revised manuscript)

2. There is no variable on reimbursement of health expenditure in consumption schedule. Hence it is difficult to say whether the expd reported by household is OOPE or total household expenditure

Response: You are right CES doesn't capture reimbursement on health expenditure. However, given the miniscule size of insurance coverage or any financial re-imburement in India until 2011-12, this does not affect our analysis in any significant way. Moreover, since our paper focuses on implication of OOP on living status, re-imburement will only help in consumption smoothening in the future whereas the OOP has impacted the living status immediately.

3. The definition of CHS used in the paper is not appropriate. Because, we know the limitation of such method in underestimating the CHS of poor. Any expd on health to those below poverty line is catastrophic.

Response: We agree that definition of CHS is matter of debate. We have employed standard definition for thresholds for analysis of catastrophe as mentioned in literature on CHS. We believe that our strategy to use range of thresholds to define catastrophe may help address the query of the reviewer. (Page 5-6 of revised manuscript)

4. State differentials or by characteristics of households are not analysed

Response: This was not the focus of our paper.

5. Need to mention whether the descriptive and CHS is based on total sample or on those who spend on health. A significant hhd did not spend any money on health care.

Response: Our estimates on CHS are based on the total sample. This has been now clarified in the manuscript.

6. Catastrophic Payments and Impoverishment due to Out-of-Pocket Health Spending by Soumita Ghosh appeared in EPW 2011 dd for 1993-94 and 2004. This need to be different from beyond expending by 2011-12. NOVEMBER 19, 2011 vol xli no 47 , EPW

Response: Our paper departs from Ghosh 2014 in one major way; we have estimated poverty and catastrophe owing to medicines. Also, we have included dimension of disease-specific estimates and inpatient and outpatient health care spending. (Page 7 of revised manuscript)

7. The word total OOPS is confusing. Should it not be used as average household health spending

Response: We have changed the terminology from 'total OOP' to 'mean OOP' in the manuscript.

8. The same way , should it not be used as MPCE at current prices,

Response: We are unable to understand this comment, please clarify.

VERSION 2 – REVIEW

REVIEWER	Sanjay K Mohanty IIPS, Mumbai
REVIEW RETURNED	06-Dec-2017

GENERAL COMMENTS	My earlier comments have been addressed
---

REVIEWER	Kiran Raj Pandey Center for Health and Social Sciences University of Chicago USA
REVIEW RETURNED	12-Dec-2017

GENERAL COMMENTS	This paper has seen significant improvement from its earlier iteration. It is now more clear and polished. A few smaller issues. Abstract conclusion: Strengthening government intervention in providing medicines free in public health care facilities has the potential to considerably reduce medicines related spending as well as total OOP payments of households As opposed to primarily focusing on what government intervention could do, authors may be better served to first conclude the facts that their methods and results have established in the abstract, that OOPs were burdensome. Within the narrow remit of the abstract conclusion, and in the absence of context, stating that government intervention could alleviate the aforementioned burden, not only appears to be a logical leap of faith but also glosses over the complexity of the issue; many of the issues that the authors have later discussed. Page 4, Line 49-52: Using 2004 NSSO data, another study highlighted that
---

hospitalization with CVD resulted in 12% higher odds of incurring catastrophic spending and 37% greater odds of falling into poverty.

May want to mention what the non-event comparator is while reporting odds (appears to be communicable diseases, on reading the referenced paper). Not doing so makes the odds look like probability, obviously not the same thing.

For cancer, the impact was greatest with the odds of catastrophic expenditures 170% higher than the odds of incurring catastrophic spending when hospital stays are due to a communicable disease condition

This sentence does it right.

Page 6 Line 49:

CES, OOP spending in NSSO 2014

Authors probably may want to stick with HMS (or NSSO 2014 HMS) instead of NSSO 2014. Since both the surveys are conducted by NSSO, it is confusing. HMS has been referred as such earlier in the text and calling it something else later throws readers off.

Page 7 Line 26-27

Estimates for both OOP payments as well as OOP payments underlying medicines expenditure by households are presented

Unclear what the authors are trying to say here. May be they mean total OOP payments as well as medicine OOP payment? Sticking to uniform terms and phrases vastly improves clarity.

Table 1 Row text:

Total household consumption expenditure

This text in this row and the row below are misleading. The numbers in the row are monthly per capita expenditures but the row text makes it look like total household expenditures, obviously not the same thing.

Page 13 Line 52

The impoverishment burden is largely driven by households spending on medicines, which accounted for over three-fourth of all medical impoverishment in India

This is a major point to stress. This has been the consistent finding from all similar papers in India and medicine related expenditures need to be targeted to prevented financial burden, whereas most government insurance programs tend to focus on inpatient stays. Especially with the onslaught of chronic diseases like heart diseases and diabetes this is only going to get worse. The other reason why financial burden due to inpatient costs is lower as compared to medicines is because, hospitalization/bed charges in India are comparatively low, and often subsidized in the public sector.

Page 14 Line 19

However, the national average masks significant underspending on medicines by several state governments, with many reportedly spending less than five percent of their health budgets.

A great follow up paper would be whether there is a difference among states in catastrophe and impoverishment due to medicines based on how much the state governments spend in medicines, or what sort of programs they have to increase the availability of medicines. Would help understand the effectiveness of government programs. For example is there a difference in financial burden between Tamil Nadu/ Rajasthan and states that don't have any programs to manage drug supplies, or spend little money on free drugs?

Page 17, Line 7

A recent pronouncement by the government intends to bring legislation for physicians to prescribe drugs only in generic names, holds even greater promise for reducing households' OOP payments on medicines and ultimately providing financial risk protection

Unsure if this holds great promise. There are some advantages sure, but there are numerous pitfalls as well. It appears that the policy will move the locus of decision making from the physician to the one who will dispense the medicine. Central to this policy's success is the assumption that the person who will dispense the medicine will exercise better judgment and be more ethical than the physician. Whether that is true is anyone's guess; I would think not.

Two methodological issues:

1. Poverty lines in India, even state specific lines, are distinguished between rural and urban, because spending patterns and needs are different between the two areas. 1. Poverty figures are also distinguished between rural and urban areas. However the authors have made no mention of this difference, even as the two populations will have clearly different consumption patterns and this difference will affect the averaged results. Was there a reason the authors chose not to mention or discuss this difference at all? Also, It would be useful to present somewhere, may be in the methods or the footnotes of table 3 what the state specific poverty lines were (just the range would do); also what the \$1.90 PPP poverty line converts to in Indian currency, so readers have an idea.

2. It appears that the inflation adjustment method used could affect comparison of consumer expenditures over the years. When I applied an inflation adjuster for India, Rs 794 in Dec 1999 was Rs 1650 in Dec 2011 prices. Is 1650 the average consumer expenditure for 2011-12 that the authors calculated? The NSSO 2011-12 CES report disaggregates its results by urban and rural areas and their average MPCE for 2011-2012 is about 1430 for rural and about Rs 2630 for urban areas. Do these figures when averaged, give Rs 1650? These figures also demonstrate how easily average figures can be skewed by the rural and urban difference.

NB: During the review process, line numbers that run continuously through the document are more helpful than ones that run for the length of the page.

VERSION 2 – AUTHOR RESPONSE

Point wise reply to the reviewers comments as follows:

This paper has seen significant improvement from its earlier iteration. It is now more clear and polished.

Response : Thanks for appreciating the revisions. We also provide point-wise rebuttal of the additional issues raised by the reviewer.

A few smaller issues.

Abstract conclusion:

Strengthening government intervention in providing medicines free in public health care facilities has the potential to considerably reduce medicines related spending as well as total OOP payments of households

As opposed to primarily focusing on what government intervention could do, authors may be better served to first conclude the facts that their methods and results have established in the abstract, that OOPs were burdensome. Within the narrow remit of the abstract conclusion, and in the absence of context, stating that government intervention could alleviate the aforementioned burden, not only appears to be a logical leap of faith but also glosses over the complexity of the issue; many of the issues that the authors have later discussed.

Response : Added “Purchase of medicines constitute the single largest component of the total OOP payments by households. Hence...”

Page 4, Line 49-52:

Using 2004 NSSO data, another study highlighted that hospitalization with CVD resulted in 12% higher odds of incurring catastrophic spending and 37% greater odds of falling into poverty.

May want to mention what the non-event comparator is while reporting odds (appears to be communicable diseases, on reading the referenced paper). Not doing so makes the odds look like probability, obviously not the same thing.

Response : Added the comparator “compared to those hospitalized with communicable conditions”

For cancer, the impact was greatest with the odds of catastrophic expenditures 170% higher than the odds of incurring catastrophic spending when hospital stays are due to a communicable disease condition

This sentence does it right.

Response : Thanks.

Page 6 Line 49:

CES, OOP spending in NSSO 2014

Authors probably may want to stick with HMS (or NSSO 2014 HMS) instead of NSSO 2014. Since both the surveys are conducted by NSSO, it is confusing. HMS has been referred as such earlier in the text and calling it something else later throws readers off.

Response : Revised as suggested

Page 7 Line 26-27

Estimates for both OOP payments as well as OOP payments underlying medicines expenditure by households are presented

Unclear what the authors are trying to say here. May be they mean total OOP payments as well as medicine OOP payment? Sticking to uniform terms and phrases vastly improves clarity.

Response : Revised as suggested

Table 1 Row text:

Total household consumption expenditure

This text in this row and the row below are misleading. The numbers in the row are monthly per capita expenditures but the row text makes it look like total household expenditures, obviously not the same thing.

Response : Revised as suggested. We have taken out the word "Total". Monthly per capita is already mentioned in the column sub-headings.

Page 13 Line 52

The impoverishment burden is largely driven by households spending on medicines, which accounted for over three-fourth of all medical impoverishment in India

This is a major point to stress. This has been the consistent finding from all similar papers in India and medicine related expenditures need to be targeted to prevented financial burden, whereas most government insurance programs tend to focus on inpatient stays. Especially with the onslaught of chronic diseases like heart diseases and diabetes this is only going to get worse.

Response : Thanks for taking note of this point. This is precisely the point we wanted to prove and this is coming up very clearly in our analysis. A number of chronic conditions (such as diabetes, joint pains etc.) require long term outpatient treatment and households end up with spending a lot of OOP on purchase of medicines for treating these conditions. We have already vividly clarified this in the discussion section.

The other reason why financial burden due to inpatient costs is lower as compared to medicines is because, hospitalization/bed charges in India are comparatively low, and often subsidized in the public sector.

Response : Although the present paper doesn't convincingly take up this analysis, we have mentioned this point in the discussion section. In the result section we don't want to suggest any reasoning/rationale.

Page 14 Line 19

However, the national average masks significant underspending on medicines by several state governments, with many reportedly spending less than five percent of their health budgets.

A great follow up paper would be whether there is a difference among states in catastrophe and impoverishment due to medicines based on how much the state governments spend in medicines, or what sort of programs they have to increase the availability of medicines. Would help understand the effectiveness of government programs. For example is there a difference in financial burden between Tamil Nadu/ Rajasthan and states that don't have any programs to manage drug supplies, or spend little money on free drugs?

Response : We have now mentioned in the discussion section that this is potential research areas for future.

Page 17, Line 7

A recent pronouncement by the government intends to bring legislation for physicians to prescribe drugs only in generic names, holds even greater promise for reducing households' OOP payments on medicines and ultimately providing financial risk protection

Unsure if this holds great promise. There are some advantages sure, but there are numerous pitfalls as well. It appears that the policy will move the locus of decision making from the physician to the one who will dispense the medicine. Central to this policy's success is the assumption that the person who will dispense the medicine will exercise better judgment and be more ethical than the physician. Whether that is true is anyone's guess; I would think not.

Response : This is a debateable policy area. We also recognize limitations of this policy initiative. Hence, we have revised the sentence and now use the phrase "holds promise for reducing households OOP".

Two methodological issues:

1. Poverty lines in India, even state specific lines, are distinguished between rural and urban, because spending patterns and needs are different between the two areas. 1. Poverty figures are also distinguished between rural and urban areas. However the authors have made no mention of this difference, even as the two populations will have clearly different consumption patterns and this difference will affect the averaged results. Was there a reason the authors chose not to mention or discuss this difference at all? Also, It would be useful to present somewhere, may be in the methods or the footnotes of table 3 what the state specific poverty lines were (just the range would do); also what the \$1.90 PPP poverty line converts to in Indian currency, so readers have an idea.

Response : We have mentioned in the method section that we have applied state specific rural and urban poverty lines. The Tendulkar Committee report provides these dis-aggregated numbers. In the present paper we are reporting only All India level average, although the data analysis were taken up at the disaggregated level by considering rural and urban poverty lines separately. Presenting rural and urban poverty estimates is not the focus of the present paper and hence we limited our results at the All India level. Rural-urban differentials in almost all the outcome indicators again may be another potential paper.

2. It appears that the inflation adjustment method used could affect comparison of consumer expenditures over the years. When I applied an inflation adjuster for India, Rs 794 in Dec 1999 was Rs 1650 in Dec 2011 prices. Is 1650 the average consumer expenditure for 2011-12 that the authors calculated? The NSSO 2011-12 CES report disaggregates its results by urban and rural areas and their average MPCE for 2011-2012 is about 1430 for rural and about Rs 2630 for urban areas. Do these figures when averaged, give Rs 1650? These figures also demonstrate how easily average figures can be skewed by the rural and urban difference.

Response : We have used state specific rural and urban price indices. For rural areas we considered consumer price index for rural labour (CPI-RL) and for urban areas consumer price index for industrial

workers (CPI-IW) were used for each state separately. Since our analysis used individual level records from the NSSO all the averages at the All India level are weighted by rural-urban and state level differentials.

NB: During the review process, line numbers that run continuously through the document are more helpful than ones that run for the length of the page.

Response : The actual pagination and line numbers are generated only after uploading the document on the journal web-site. We revise the document in a word file which will have different line numbers and may be confusing for reviewer/s.